# Breast Cancer in Patients with Previous Endometriosis Showed Low Aggressive Subtype

**DOI:** 10.3390/medicina60040625

**Published:** 2024-04-12

**Authors:** Gianluca Vanni, Aikaterini Selntigia, Valentina Enrica Marsella, Consuelo Russo, Marco Pellicciaro, Marco Materazzo, Giuseppe Rizzo, Oreste Claudio Buonomo, Caterina Exacoustos

**Affiliations:** 1Breast Unit, Department of Surgical Science, University of Rome “Tor Vergata”, Viale Oxford 81, 00133 Rome, Italy; vanni_gianluca@yahoo.it (G.V.); marco.pellicciaro@ptvonline.it (M.P.); marco.materazzo@alumni.uniroma2.eu (M.M.); o.buonomo@inwind.it (O.C.B.); 2Obstetrics and Gynecological Unit, Department of Surgical Science, University of Rome “Tor Vergata”, Viale Oxford 81, 00133 Rome, Italy; aikateriniselntigia@gmail.com (A.S.); crconsuelorusso@gmail.com (C.R.); caterinaexacoustos@tiscali.it (C.E.); 3Ph.D. Program in Translation Medicine, Department Biomedicine and Prevention, University of Rome “Tor Vergata”, Viale Oxford 81, 00133 Rome, Italy; 4Ph.D. Program in Applied Medical-Surgical Sciences, Department of Surgical Science, University of Rome “Tor Vergata”, Viale Oxford 81, 00133 Rome, Italy; 5Obstetrics and Gynecological Unit, Department Biomedicine and Prevention, University of Rome “Tor Vergata”, Viale Oxford 81, 00133 Rome, Italy; giuseppe.rizzo@ptvonline.it; 6General Surgery Program, UNIBAS, University of Basilicata, Via dell’Ateneo Lucano, 10, 85100 Potenza, Italy

**Keywords:** breast neoplasm, endometriosis, breast cancer subtype, malignant neoplasm of breast, endocrine breast disease, breast disease, immunophenotyping, triple negative breast cancer, infertility female, reproduction

## Abstract

*Background and Objectives*: The association between endometriosis and breast cancer still remains controversial. The aim of this study was to investigate the different subtypes of breast cancer, immunohistochemical markers, hormone receptors, and ki67 proliferation indexes in patients with and without endometriosis and/or adenomyosis. *Materials and Methods*: All patients with endometriosis and breast cancer were enrolled. Women with endometriosis and breast cancer (Group BC+EN+) were compared to patients with breast cancer without endometriosis (group BC+EN−) and those with endometriosis without breast cancer (group BC-EN+). General population characteristics and histological and immunohistochemical subtypes of breast cancer were compared between groups. *Results*: Our study included 41 cases affected by both endometriosis and/or adenomyosis and breast cancer (Group BC+EN+) that were matched (1:2) with 82 patients affected only by breast cancer (group BC+EN−) and 82 patients affected only by endometriosis and/or adenomyosis (group BC-EN+). Group BC+EN+ presented a higher percentage of ER receptor expression (83% vs. 70%, *p* = 0.02), as well as lower values of Ki 67% (15% vs. 24%, *p* < 0.0001) and HER2+ (9.8% vs. 28%, *p* = 0.022). These findings were more evident when comparing patients with premenopausal status, while in postmenopausal patients, this difference was no longer significant. Regarding endometriosis, no statistical differences were observed in type or specific localization of the disease among the groups with and without breast cancer. *Conclusions*: Patients with endometriosis presented lower aggressive breast cancer rates with higher values of ER% and lower values of Ki 67 and HER2neu+. The type and severity of endometriotic diseases seemed not to influence breast cancer occurrence.

## 1. Introduction

Endometriosis is a chronic benign gynecological disease that has been widely investigated due to its high prevalence, estimated at 10–20% of women of reproductive age and 35–50% of infertile women [1,2]. Although endometriosis is a benign disease, special attention has been paid to the similar behavioral pattern to some cancers, as they both exhibit uncontrolled, estrogen-dependent proliferation, invasion, neo-angiogenesis, and metastases [3,4].

Over the years, several studies have established an association between endometriosis and certain types of malignancies, particularly ovarian cancer [5,6,7,8], cutaneous melanoma, and non-Hodgkin’s lymphoma [9]. However, the association between endometriosis and other hormone-dependent tumors such as breast cancer and endometrial cancer still remains controversial [10]. Currently, particular scientific interest is focused on the connection between endometriosis and breast cancer, since both diseases have high prevalence and common risk factors associated with abnormal hormonal activity and inflammatory and environmental factors, such as early menarche age, late menopause age, nulliparity, and infertility [11].

The literature is heterogeneous regarding the correlation between endometriosis and breast cancer: Many studies suggest an association between the two diseases [11,12,13,14,15,16,17,18,19,20,21,22,23,24,25,26,27], of which only five show a significant correlation [12,21,25,26,27]. Several studies claim that a positive history of endometriosis decreases the risk of developing breast cancer [15,21,28,29,30], while others demonstrate that the two diseases are not associated [11,13,14,16,17,31]. Additionally, many of the treatments for and complications of endometriosis, including oral contraceptives (e.g., progestin, estro-progestin, GnRH agonist), analgesic use, oophorectomy, and infertility, may influence breast cancer risk [32]. In particular, regarding hormone use, Bertelsen et al. suggested that the use of GnRH agonists in women with endometriosis might have a protective effect against breast cancer [17]. Further, Dahila et al. reported a significantly lower use of oral contraceptives in patients with breast cancer [27]. Both treatments are used in women with endometriosis, especially prolonged estro-progestin therapy, which could constitute a risk factor for breast cancer. The risk factor relationship in breast cancer is affected by tumor hormone receptors and menopausal status at time of diagnosis [11]. In the literature, there are only two studies which have investigated the correlation between tumor hormone receptors and endometriosis [11,14]. While biannual BC mammographic screening remains the cornerstone of a BC screening program to reduce mortality and side effects related to the treatment, several authors have tried to underline the role of EN on BC risk factor development to plan a tailored screening program for these patients [11,27,33].

However, to our knowledge, no previous studies have analyzed the association between endometriosis and breast cancer type or immunohistochemical biomarkers. It is important to highlight that, in all studies conducted to investigate this correlation, the diagnosis of endometriosis was based on surgical detection, which certainly underestimates the prevalence of the disease. In fact, a noninvasive diagnosis of endometriosis by imaging also included patients who never underwent surgery, but still exhibited signs of pelvic disease. Since the publication of the 2022 ESHRE guidelines 12, imaging is recognized as a diagnostic tool, without the need for surgical or histological confirmation, in particular for endometriomas, deep endometriosis, and adenomyosis [34].

The main purpose of this study was to investigate the different subtypes of breast cancer in patients with endometriosis. Notably, the different characteristics of breast cancer, such as immunohistochemical biomarkers, hormone receptors, cancer type, and menopausal status at diagnosis, were compared among patients with and without endometriosis. Furthermore, adenomyosis as part of the same disease and other endometriosis types were also evaluated in patients with and without breast cancer.

## 2. Materials and Methods

### 2.1. Study Design and Study Participants

A double 1:2 matched-pair analysis was performed from two prospectively maintained databases (breast cancer database and endometriosis database) of the Gynecological Unit Endometriosis Center and the Breast Unit of University Hospital ‘Policlinico Tor Vergata’ of Rome, respectively. The study was conducted according to institutionally approved protocols at each study site, and informed consent to data use for research was provided by all patients at first access to both units. All patients with endometriosis and/or adenomyosis (EN+) diagnosed through surgery or transvaginal ultrasound (TVUS) who were also affected by breast cancer (BC+) were enrolled and compared to patients with breast cancer (BC) without endometriosis (BC+EN−) and to patients with endometriosis and/or adenomyosis (EN) without BC (BC-EN+).

The inclusion criteria for the study group population (Group BC+EN+) were women aged 20–70 years old with:-History of with endometriosis and/or adenomyosis diagnosed through surgery or ultrasonographic examination in our endometriosis center who subsequently received a new BC diagnosis;-Histological diagnosis of invasive breast cancer on a surgical specimen.

The exclusion criteria were incomplete clinical, histological, and anamnestic data; other oncological disease; pregnancy; neoadjuvant therapy; and absence of consent to the processing of personal data. For each case of breast cancer and concomitant diagnosis of endometriosis, controls with a diagnosis of breast cancer without endometriosis (Group BC+EN−) and a diagnosis of endometriosis without breast cancer (Group BC-EN+) were retrospectively randomly enrolled by querying the electronic database of both units. Control groups were matched for age (±5 years), BMI (±1), and menopausal status.

All patients diagnosed with breast cancer underwent TVUS at our endometriosis center to rule out endometriosis and other gynecological diseases before starting medical treatment. All endometriosis patients without breast cancer underwent a breast examination in our Breast Unit or followed the recommended screening programs with ultrasound and/or mammography check-ups, depending on their age.

### 2.2. Clinical Evaluation

Patient data were recorded according to a pre-established format using the Filemaker pro^®^ software version 9.0 (Filemaker Inc., Santa Clara, CA, USA). For each patient, a detailed history was collected regarding personal medical, clinical, and obstetric history. The data included date of birth, body mass index (BMI; kg/m^2^), age at menarche, menstrual cycle characteristics, last menstrual period, parity (number of all previous pregnancies: spontaneous abortions and/or live births), primary or secondary infertility, previous surgical interventions, and other diseases. Infertility was defined as no pregnancy after 12 months of unprotected intercourse [35]. 

Medications taken before breast cancer diagnosis for any reason were recorded, including hormonal and non-steroidal anti-inflammatory drugs. Namely, previous hormonal therapy, current hormonal therapy, and hormone replacement therapy (HRT) in menopause were recorded.

### 2.3. Gynecological Unit Evaluation

The diagnosis of endometriosis was confirmed through previous pelvic surgical interventions and/or a detailed ultrasonographic examination.

The surgical diagnosis of endometriosis was made based on the surgical report and the histological exam in order to register the subtype and the localization of the lesions. Cases in which the surgical report and the histological exam were not available were excluded from the study. Particularly, the localization and subtype of endometriosis in terms of endometriomas, posterior deep endometriosis, anterior deep endometriosis, and adenomyosis were collected from clinical charts.

Furthermore, sonographic examinations in patients with only ultrasonographic diagnosis of endometriosis or with residual disease after surgery were performed in our endometriosis center in order to confirm and describe the endometriotic/adenomyotic disease. To assess the type and location of the disease, we used only TVUS and not RMI, since these two imaging tools have similar diagnostic accuracy for endometriotic/adenomyotic lesions. Moreover, TVUS provides a rapid result and is cost-effective compared to MRI.

The ultrasound examination was performed using a Voluson E6 or E8 device (GE Healthcare, Zipf, Austria) with a transvaginal probe. A conventional two-dimensional (2D) ultrasound with grey scale and power Doppler for pelvic assessment was carried out, evaluating the uterus, the myometrium, and the endometrium in detail in order to exclude any abnormalities. Two-dimensional examination was followed by acquisition of three-dimensional imaging of the uterus with and without a power Doppler. Next, the scan examined the adnexa, the pouch of Douglas, and other pelvic organs (bladder, rectum, rectosigmoid junction, ureters) and sites (posterior, lateral, and anterior parametria; rectovaginal septum; vesico-uterine pouch; uterosacral ligaments). Endometriosis features were always carefully scanned using a previously published ultrasound mapping system [36,37]. The sonographic diagnosis of ovarian endometrioma was defined by the presence of a persistent unilocular or multilocular (less than five locules) cyst characterized by a homogeneous low-level echogenicity (ground glass echogenicity) of the cystic fluid and absent or moderate vascularization of the cystic walls [38]. The diagnosis of deep infiltrating endometriosis (DIE) was made if at least one structure in the anterior, posterior, or lateral compartments showed the presence of an abnormal retroperitoneal hypoechoic linear or nodular thickening with irregular contours and no or few Doppler signals, according to previously described and validated ultrasonographic criteria [36,37]. We reported the DIE localization in relation to the pelvic compartment: anterior (bladder, vesico-uterine septum), posterior (rectum, torus posterior vaginal fornix, RVS), and lateral (bilateral USL parametria, ureters). Sonographic findings of uterine adenomyosis were also recorded, and its diagnosis was made when at least one direct morphological feature was present: myometrial cysts, hyperechoic islands, or an irregular and infiltrated endometrial–myometrial junction zone on either 2D or 3D imaging [39]. These findings have been described previously, and there is a wide consensus regarding their reliability as morphological markers of adenomyosis [39]. All the scans were stored as 2D still images, 2D video-clips, and 3D volumes.

Patients were considered to be affected by endometriosis and/or adenomyosis if TVUS showed clear direct and evident ultrasonographic findings of the pelvic disease. According to the ESHRE Guidelines of 2022, these TVUS direct features can be considered diagnostic and do not need surgical confirmation [34].

### 2.4. Breast Unit Evaluation

All data from patients were retrieved from clinical notes. All patients subjected to breast surgery were evaluated with at least a triple assessment: mammography, breast ultrasonography, and biopsy. Patients with radiologically suspicious lesions underwent preoperative diagnosis, which was obtained through cytological examination or biopsy (core needle biopsy or vacuum-assisted biopsy). Data regarding pre-operative diagnosis were retrieved from radiological and pathological examination reports.

Once BC was diagnosed, a multidisciplinary treatment was planned according to the Italian guidelines published at the time, and neoadjuvant treatment was planned if required.

Surgical procedures included breast-conserving surgery (BCS) and mastectomy. BCS encompassed all procedures defined as partial mastectomy (e.g., wide local excision, quadrantectomy), while mastectomy included the complete removal of glandular tissue regardless of the sparing of skin or nipple areola complex (e.g., nipple-sparing mastectomy, skin-sparing mastectomy, and radical mastectomy).

Patients without clinical or radiological evidence or suspicion of lymph nodes metastasization underwent a sentinel lymph node biopsy. Alternatively, patients with axillary involvement underwent axillary lymph node dissection. Axillary lymph node dissection was considered the removal of at least 10 lymph nodes, according to current guidelines. Tumor maximum diameters were collected and reported in millimeters. Pathological staging was based on recommendations from AJCC 2018 (edition VIII) of TNM classification [40]. Tumor grading was evaluated according to the Nottingham Histologic Score system (the Elston–Ellis modification of the Scarff–Bloom–Richardson grading system).

Estrogen receptor (ER), progesterone receptor (PR), and Ki67 index were expressed as percentages of positive cells in specimens studied through immunohistochemistry. Overexpression of the Her2 gene (HER2+) was identified by immunohistochemistry or by FISH, as indicated by the recommendations of the 2018 ASCO/CAP, and reported as a score. All patients were classified, according to the classification of intrinsic subtypes recommended by the San Gallen International Expert Consensus Report of 2017 [41], into the following subgroups: Luminal A, Luminal B+, Luminal B-, Her2 Type (Her2), and Triple Negative BC. They were then reported in the analysis of the final pathological examination.

Following surgery, adjuvant locoregional and systemic treatment were planned according to the Italian guidelines published at the time.

### 2.5. Statistical Analysis

Statistical analysis was performed using the Statistical Package for the Social Sciences (SPSS v.15.0; SPSS, Inc., Chicago, IL, USA). For population characteristics, all continuous variables were expressed in terms of mean ± SD, while categorical variables were indicated as frequencies and percentages. General characteristics of the study population (Group BC+EN+) were initially compared with the control group (BC+EN−). These two groups were subdivided into premenopausal and postmenopausal women and compared in terms of general characteristics and breast cancer histological features. Finally, a comparison between premenopausal cases of breast cancer and endometriosis (Group BC+EN+) and endometriosis without breast cancer (Group BC-EN+) was performed, considering general characteristics s well as endometriosis location and types. Continuous variables between groups were compared with the *t*-Student test or Mann–Whitney U test, according to the Kolmogorov–Smirnov test. Categorical and dichotomous variables were compared between groups with the chi-square test or Fisher’s exact test according to the sample size, while for multiple categorical variables, Montecarlo correction was applied for both tests (e.g., T stage, biomolecular classification). All variables with *p* < 0.05 according to monovariate analysis were reported as statistically significant.

Additionally, variables with *p*-values < 0.10 were included in a logistic regression statistical analysis in order to estimate the effect on less aggressive breast cancer (defined as ki67 < 20%) using the forward Wald method. The Hosmer–Lemeshow test was applied to evaluate the goodness of fit and calibration for logistic regression models.

## 3. Results

From June 2020 to May 2023, a total of 1546 patients with BC or endometriosis were admitted to our facilities. Of these, 41 were included in our analysis (Group BC+EN+) after meeting the inclusion criteria. Of these, 21 patients had a diagnosis of endometriosis; surgery was performed in 7 patients in our Gynecological Unit and in 14 in different Italian centers. Since different classifications of endometriosis staging were used by the different surgical teams, we relied on the endometriosis type and location assessment included in the surgical and histological reports. In the study period, 843 patients received a diagnosis. A total of 234 were diagnosed with endometriosis in our endometriosis center. Among the total patients, 442 were also evaluated in our breast unit. A total of 41 patients were excluded due to a previous or new breast cancer diagnosis. Therefore, 401 cases were matched in a 2:1 design for age, BMI, and menopausal status, and eventually, 82 patients were selected for the BC-EN+ group. During the same period, 703 patients with breast cancer were admitted to our facilities, out of which 388 had undergone a gynecological and TVUS evaluation in our gynecological unit. Additionally, 41 patients were excluded due to a previous or new diagnosis of endometriosis. Therefore, out of the remaining 347 patients, 82 were matched in a 1:2 design for age, BMI, and menopausal status and were selected for the BC+EN− group. The study flowchart is displayed in Figure 1.

Consequently, the 41 patients with breast cancer and endometriosis (Group BC+EN+) fulfilling the inclusion criteria were initially compared to the 82 patients with BC without EN (Group BC+EN−). The general characteristics of these two groups were similar in terms of mean age at breast cancer diagnosis (46.5 vs. 46.1 years, *p* = 0.699) and in terms of mean age at menarche (12.1 vs. 12.3 years, *p*= 0.525). Contrastingly, comparisons showed a significant statistical difference in parity (39% vs. 16%, *p* = 0.006), infertility (27% vs. 5%, *p* = 0.001), and previous hormonal treatment (76% vs. 45%, *p* = 0.002), as reported in Table 1. Then, the two groups were compared according to menopausal status by dividing patients into premenopausal and postmenopausal subgroups. The general characteristics of premenopausal and postmenopausal patients showed similar statistical results in terms of mean age at breast cancer diagnosis, BMI, and mean age at menarche, thus confirming our trend (Table 1).

All histological characteristics among the two groups with BC were compared and exhibited significant differences, with a higher value of ER receptor expression in BC+EN+ patients (83.2 ± 21.2 vs. 70.1 ± 34.0, *p* = 0.010), as shown in Table 2. Moreover, Group BC+EN+ patients presented a lower proliferation index (Ki67%) (14.8 ± 10.4 vs. 24.4 ± 18.7 *p* < 0.0001) as well as a significantly lower expression of HER2 compared with Group BC+EN− patients (9.8% vs. 28%, *p* = 0.022). An additional significant difference was a higher rate of sentinel lymph node biopsy positivity in Group BC+EN− patients (9.8% vs. 47.6%, *p* < 0.001). Comparing the tumor sizes in the two patient cohorts, the BC+EN− group had larger average tumors than the BC+EN+ group (1.4 vs. 2.0), with a relative *p*-value of 0.043.

Upon the Montecarlo test, comparing immunohistochemical subtypes, a significant difference was reported with *p* value = 0.017. The Luminal A subtype, representing the least aggressive subtypes of breast cancer, was more represented in Group BC+EN+ (73.2% vs. 40.2%). Contrastingly, the Luminal B HER2+ subtype, which is biologically more aggressive, was more represented in Group BC+EN− (23.2% vs. 9.8%).

Moreover, the Montecarlo test comparing T staging and N staging showed both to be significantly different, with a *p* value < 0.001.

Lastly, the Montecarlo test comparing grading showed a significant difference with a *p* value < 0.001. Group BC+EN− presented more G3 tumors than Group BC+EN+ (36.6% vs. 7.3%). Other results and relative *p* values are shown in Table 2.

In our study group, only 26% (11/41) of patients were in the postmenopausal period; therefore, we concentrated our initial comparison on premenopausal patients, in which both diseases are generally more aggressive. Considering only patients with premenopausal status, Group BC+EN+ patients had smaller tumors than BC+EN− (1.5 vs. 2.1), with a *p* value 0.049. Moreover, Group BC+EN+ patients exhibited a higher value of ER receptor expression (82.5 ± 24.2 vs. 69.3 ± 32.9, *p* value = 0.035). This group also showed a lower expression of HER2 (33.3% vs. 6.7%, *p* value = 0.001) and a lower proliferation index (Ki67%) (14.4 ± 10.1 vs. 26.4 ± 19.6, *p* value < 0.001). Patients in group BC+EN− had more positive lymph nodes at sentinel node biopsy than group BC+EN+ (6.7% vs. 51.7%, *p* value < 0.001). Comparing immunohistochemical subtypes between the two groups in the Montecarlo test, a significant difference was found, with a *p* value = 0.018. Also, in this case, the Luminal A subtype was more represented among Group BC+EN+ patients (73.3% vs. 38.3%), while the Luminal B HER2+ subtype was more represented among Group BC+EN− patients (28.3% vs. 6.7%).

Moreover, significant differences were found in the Montecarlo test in terms of T staging and N staging, *p* = 0.042 and *p* < 0.001, respectively.

Comparing grading statuses in the Montecarlo test, an important difference was found with a *p* value < 0.001. Patients in Group BC+EN− presented with more G3 tumors (41.7% vs. 10%), while patients in Group BC+EN+ exhibited more G1 tumors (50% vs. 13.3%).

Conversely, considering patients with postmenopausal status, no differences between the two groups were found in terms of histological characteristics or immunohistochemical subtypes. Other results of postmenopausal BC characteristics between the two groups are reported in Table 2.

At univariate logistic regression, age of breast cancer diagnosis and menopause age were found to be predictive factors for Ki67 < 20%, *p* = 0.005 (OR = 0.917; 95%CI: 0.862–0.974) and *p* = 0.028 (OR = 0.926; 95%CI: 0.865–0.992), respectively. Endometriosis, at univariate logistic regression, was found to be a predictive factor for BC, with Ki67% < 20%: *p* = 0.050 (OR = 0.446; 95%CI: 0.197–1.009). Lastly, menopausal status at time of diagnosis, at univariate logistic regression, was found to be a predictive factor for BC, with Ki67% < 20%: *p* = 0.100 (OR = 0.490; 95%CI: 0.205–1.173).

During multivariate logistic regression, we considered age at breast cancer diagnosis, menopause, endometriosis, and hormonal treatment as predictive factors of breast cancer with Ki67 < 20%. Age at breast cancer diagnosis and the presence of endometriosis were found to be independent predictive factors, *p* = 0.040 (OR = 0.816; 95%CI: 0.671–0.991) *p* = 0.012 (OR = 3.203; 95%CI: 1.285–7.986), respectively. At multivariate logistic regression analysis, which was carried out in order to predict breast cancer with a low Ki67 index inferior to 20%, only endometriosis was a significant predictive factor for cancer with a lower Ki67 index, *p* = 0.010 (OR = 3.3882; 95%CI: 1.354–11.135). Other parameters considered at multivariate logistic regression and relative *p* and OR values are shown in Table 3.

Contrastingly, at univariate logistic regression, considering postmenopausal patients with diagnoses of breast cancer, endometriosis was not predictive of breast cancer with a lower Ki67 index; *p* = 1.000 (OR = 1.000; 95%CI: 0.197–5.079). At multivariate logistic regression, none of the factors of the previous model were significant predictors of a low proliferation index (Table 3). 

Furthermore, we analyzed the different types of endometriosis in patients with other comorbidities (30 patients) compared to those with only endometriosis (60 patients) in the premenopausal age group. We demonstrated that the presence of deep infiltrating posterior endometriosis and adenomyosis was significantly lower in the group with other comorbidities compared to the group with only endometriosis (57% vs. 85%, *p* = 0.003 and 43% vs. 65% *p* = 0.048, respectively) (Table 4).

## 4. Discussion

Many studies have attempted to understand the correlation between endometriosis and the development of neoplastic diseases by aiming to show how this gynecological condition increases the risk of cancers, including breast cancer [12,17,27]. Endometriosis and up to 90% of breast cancer phenotypes are hormone-related diseases which exhibit uncontrolled, estrogen-dependent cellular proliferation, neo-angiogenesis, invasion, and metastases [42]. Therefore, in our study, we attempted to highlight a correlation between BC and endometriosis, resulting in a pathogenetic association between low-risk hormonal BC and endometriosis in premenopausal populations.

BC in premenopausal women, while accounting for 7–10% of all BC diagnoses, is the leading oncological diagnosis associated with insufficient or unsatisfactory therapeutic options [43]. Specifically, when premenopausal BC patients are compared with the general BC population, they are faced with peculiar characteristics regarding tumor biology and psychosocial aspects [44]. In fact, low-risk hormonal BC is usually less common in younger populations, which tend to exhibit higher rates of aggressive subtypes due to the high penetrance of breast cancer susceptibility genes and/or BC family history [45]. Additionally, some modifiable risk factors, such as BMI, have been associated with specific BC subtypes [46]. A growing body of evidence demonstrates how chronic subclinical inflammation plays a pivotal role in BC and tumor progression locally and systematically [47].

Women with endometriosis have impaired immune function, which can cause upregulation of local estrogens. Similarly, mediators of inflammation, particularly IL-6, upregulate estrogen levels in the breast by inducing aromatase activity [26]. However, evidence linking endometriosis with BC is rather vague and relies on the hormonal dependence and common clinical characteristics of both diseases [14,29] such as early age at menarche, late menopause, and nulliparity [48]. Many studies have attempted to find a correlation between BC and endometriosis with contradictory results. Only five reported statistically significant correlations between BC and endometriosis [12,21,25,26,27]. Conversely, in a large study published in 2016, Farland et al. did not report a clearly increased risk of developing breast cancer in patients with endometriosis, but a higher rate of ER+PR− BC was demonstrated [11].

Additionally, in a recent analysis carried out by Gemke et al., no statistically significant differences were found between endometriosis and subsequent BC at a median follow-up of 10 years. However, while the sample size and follow-up period were sufficient, the lack of immunohistochemical classification did not allow a higher rate of luminal subtypes to be reported following an analysis between groups [31].

Though smaller in sample size and lacking a follow-up period, the data collected in our study seem to confirm the association between endometriosis and some BC subtypes with higher expression of ER in the BC+EN+ group when compared with the BC+EN− group. In our opinion, endocrine imbalance may be responsible for ectopic endometrial proliferation and hormone-sensitive BC progression.

As expected, in our study, a statistically significantly lower rate of the HER2 phenotype was reported in Group BC+EN+ patients, providing additional insight for a specific hormone-sensitive carcinogenic pathway in patients with endometriosis.

Contrary to our expectations, our results were not confirmed in postmenopausal women, where the rates of different molecular subtypes were not different between the postmenopausal groups. This apparent lack of correlation may be attributed to the higher rate of hormone-positive diagnosis in the postmenopausal patient groups, or to the different BC risk factors linked with this subset of the population in comparison with younger populations [49]. Critically, premenopausal Group BC+EN+ showed a higher rate of Luminal A when compared with premenopausal Group BC+EN−, which is considered an indolent BC phenotype with the lowest risk of recurrence. Luminal A usually exhibits low Ki67, negative HER2, and a low nuclear grading. Finally, locoregional spreading in the axillary lymphatic tissue, the strongest known risk factor for distant relapse, confirmed our hypothesis, with a higher rate of lymph node involvement in patients without endometriosis [50,51,52,53,54].

We are aware that our study may have some limitations. Firstly, the retrospective monocentric analysis may have influenced our results. However, our data were collected from two prospectively maintained databases, and the implemented monocentric design may have strengthened the internal validity of the study. Additionally, due to the small sample size of the study, no power analysis was performed to detect statistical significance, but the matched paired analysis may have reduced the risk of bias. Moreover, the lack of follow-up may reduce the strength of our results. While some authors may argue that specific hormonal impairment may play a role in long-term follow-up in these subsets of patients (EN+BC+), our data did not investigate long-term follow-up, and further studies are needed in order to assess the long-term outcomes of endometriosis and BC. Additionally, patients undergoing neoadjuvant chemotherapy were excluded from the analysis. These exclusion criteria were added due to the risk of immunophenotypic changes related to the neoadjuvant treatment. Further studies by our group will specifically focus on locally advanced BC, neoadjuvant chemotherapy, and EN.

## 5. Conclusions

Our study demonstrates how premenopausal women with endometriosis and breast cancer exhibit higher rates of low-risk hormonal BC, requiring multidisciplinary treatment that strives for quality of life improvement and fertility preservation, in light of the potentially better clinical outcomes when compared with the general population.

## Figures and Tables

**Figure 1 medicina-60-00625-f001:**
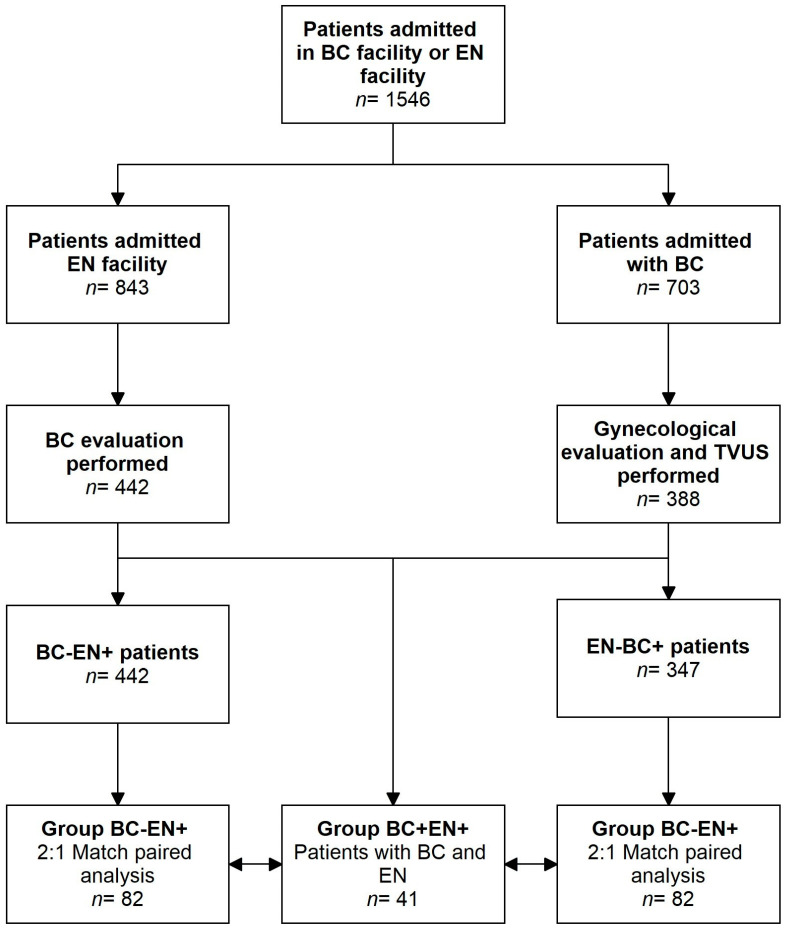
Study Flowchart. BC: breast cancer; EN: endometriosis; TVUS: transvaginal ultrasound.

**Table 1 medicina-60-00625-t001:** Demographic characteristics in the study population group with breast cancer and endometriosis (BC+EN+) versus the group with breast cancer without endometriosis (BC+EN−), according to the menopausal status.

	Overall Population (*n*= 123)			Premenopausal Group (*n=* 90)			Postmenopausal Group (*n=* 33)		
	BC+ EN+ Group (*n =* 41)	BC+ EN- Group (*n =* 82)	*p*-Value	BC+ EN+ Group (*n =* 30)	BC+ EN- Group (*n =* 60)	*p*-Value	BC+ EN+ Group (*n =* 11)	BC+ EN- Group (*n =* 22)	*p*-Value
Age at breast cancer diagnosis (y) mean ± SD	46.5 ± 7.3	46.1 ± 7.5	0.699	42.7 ± 5.0	43.4 ± 4.7	0.699	53.5 ± 6.6	54.6 ± 6.8	0.641
BMI (kg/m^2^) mean ± SD	21.4 ± 3.0	22.2 ± 3	0.525	22.3 ± 3.0	23.0 ± 4.0	0.521	22.0 ± 2.0	23.0 ± 3.1	0.116
Age at menarche (yrs) mean ± SD	12.1 ± 1.2	12.3 ± 1.5	0.310	12.1 ± 1.0	12.2 ± 1.2	0.627	12.0 ± 1.1	12.2 ± 1.2	0.984
Previous hormonal treatment n (%)	31 (76.0%)	37 (45.0%)	0.002 *	14 (48.0%)	1 (2.0%)	<0.001 *	7 (64.0%)	2 (9.0%)	0.002 *
Pelvic surgery n (%)	21 (53.0%)	3 (4.0%)	<0.001 *	9 (30.0%)	4 (7.0%)	0.001 *	2 (18.0%)	0 (0%)	0.104
Infertility n (%)	11 (27.0%)	4 (5.0%)	0.001 *	14 (47.0%)	10 (17.0%)	0.004 *	2 (18.0%)	3 (14.0%)	1.000
Nulliparous n (%)	16 (39.0%)	13 (16.0%)	0.006 *	14 (47.0%)	10 (17.0%)	0.004 *	2 (18.0%)	3 (14.0%)	1.000

All continuous variables are expressed in terms of mean ± standard deviation (SD), while categorical variables are indicated as frequencies and percentages. Continuous variables between groups were compared with *t*-Student test or Mann–Whitney U test, according to the Kolmogorov–Smirnov test. Categorical and dichotomous variables were compared between groups with chi-square test or Fisher’s exact test according to the sample size. *p*-values < 0.05 are highlighted with * and considered statistically significant. BC: breast cancer; EN: endometriosis; BMI: body mass index.

**Table 2 medicina-60-00625-t002:** Clinicopathological and immunohistochemical breast cancer (BC) characteristics in the study population group with breast cancer and endometriosis (BC+EN+) versus the group with breast cancer without endometriosis (BC+EN−), and sub-analysis according to the menopausal status.

	Overall Population (*n* = 123)			Premenopausal Group (*n* = 90)			Postmenopausal Group (*n* = 33)		
	BC+ EN+ Group (*n* = 41)	BC+ EN− Group (*n* = 82)	*p*-Value	BC+ EN+ Group (*n* = 30)	BC+ EN− Group (*n* = 60)	*p*-Value	BC+ EN+ Group (*n* = 11)	BC+ EN− Group (*n* = 22)	*p*-Value
Dimension (cm) mean ± SD	1.4 ± 1.2	2.0 ± 1.5	0.043 *	1.5 ± 1.2	2.1 ± 1.7	0.049 *	1.3 ± 1.0	1.6 ± 1.0	0.323
Macrometastatic SLNB n (%)	4 (9.8%)	39 (47.6%)	<0.001 *	2 (6.7%)	31 (51.7%)	<0.001 *	2 (18.2%)	8 (36.4%)	0.430
pT n(%)									
pT1a	5 (12.2%)	5 (6.1%)	0.036 *	3 (10.0%)	3 (5.0%)	0.042 *	2 (9.1%)	2 (18.2%)	0.386
pT1b	14 (11.4%)	11 (8.9%)		11 (36.7%)	8 (13.3%)		3 (27.3%)	3 (13.6%)	
pT1c	12 (29.3%)	32 (39.0%)		9 (30.0%)	19 (31.7%)		3 (27.3%)	13 (59.1%)	
pT2	10 (24.4%)	31 (37.8%)		7 (23.3%)	27 (45.0%)		3 (27.3%)	4 (18.2%)	
pT3	0 (0.0%)	3 (3.7%)		0 (0.0%)	3 (5.0%)		3 (27.3%)	4 (18.2%)	
pN n (%)									
pN0	37 (90.2%)	45 (54.9%)	<0.001 *	28 (93.3%)	29 (48.3%)	<0.001 *	9 (81.8%)	16 (72.7%)	0.455
pN1a	2 (4.9%)	27 (32.9%)		1 (3.3%)	22 (36.7%)		1 (9.1%)	5 (22.7%)	
pN1b	1 (2.4%)	1 (1.2%)		0 (0.0%)	1 (1.7%)		1 (9.1%)	0 (0.0%)	
pN2a	0 (0.0%)	5 (6.1%)		0 (0.0%)	5 (8.3%)		0 (0.0%)	0 (0.0%)	
pN2b	0 (0.0%)	1 (1.2%)		0 (0.0%)	0 (0.0%)		0 (0.0%)	1 (4.5%)	
pN3	1 (2.4%)	3 (3.7%)		1 (3.3%)	3 (5.0%)		0 (0.0%)	0 (0.0%)	
Histology n (%)									
Invasive Lobular	9 (21.95%)	11 (13.4%)	0.299	7 (23.3%)	7 (11.7%)	0.216	2 (18.2%)	4 (18.2%)	1.000
Invasive Ductal	32 (78.04%)	71 (86.6%)		23 (73.3%)	53 (88.3%)		9 (81.8%)	18 (81.8%)	
Hormonal Expression n (%)									
ER+/PR−	3 (7.3%)	8 (9.8%)	0.226	3 (10%)	5 (8.3%)	0.925	0 (0.0%)	3 (13.6%)	0.555
ER+/PR+	36 (87.8%)	62 (75.6%)		25 (83.3%)	47 (78.3%)		11 (100%)	15 (68.2%)	
ER-/PR−	2 (4.9%)	12 (13.4%)		2 (6.7%)	8 (13.3%)		0 (0.0%)	4 (13.6%)	
ER % mean ± SD	83.2 ± 21.2	70.1 ± 34.0	0.010	82.5 ± 24.2	69.3 ± 32.9	0.035	85.0 ± 9.48	72.0 ± 37.4	0.139
PR % mean ± SD	54.3 ± 35.5	59.1 ± 39.2	0.505	56.2 ± 36.6	62.6 ± 39.1	0.457	49.1 ± 33.4	49.7 ± 38.7	0.961
HER 2+ n (%)	4 (9.8%)	23 (28.0%)	0.022	2 (6.7%)	20 (33.3%)	0.001	2 (18.1%)	3 (13.6%)	0.375
KI67 % mean ± SD	14.8 ± 10.4	24.4 ± 18.7	<0.001	14.4 ± 10.1	26.4 ± 19.6	<0.001	16.0 ± 11.0	18.0 ± 14.0	0.626
Immunohistochemical subtypes n (%)									
Luminal A	30 (73.2%)	33 (40.2%)	0.017	22 (73.3%)	23 (38.3%)	0.018	8 (72.7%)	10 (45.5%)	0.371
Luminal B (Her 2−)	5 (12.2%)	17 (20.7%)		4 (13.3%)	12 (20.0%)		1 (9.1%)	5 (22.7%)	
Luminal B (Her 2+)	4 (9.8%)	19 (23.2%)		2 (6.7%)	17 (28.3%)		2 (18.2%)	2 (9%)	
Her 2+	0	4 (4.9%)		0	3 (5.0%)		0	1 (4.5%)	
Triple −	2 (4.9%)	9 (11%)		2 (6.7%)	5 (8.3%)		0	4 (18.2%)	
Grading n (%)									
G1	18 (43.9%)	12 (14.6%)	<0.001	15 (50.0%)	8 (13.3%)	<0.001	3 (27.3%)	4 (18.2%)	0.288
G2	20 (48.8%)	40 (48.8%)		12 (40.0%)	27 (45.0%)		8 (72.7%)	13 (59.1%)	
G3	3 (7.3%)	30 (36.6%)		3 (10.0%)	25 (41.7%)		0	5 (22.7%)	

All continuous variables are expressed in terms of mean ± SD, while categorical variables are indicated as frequencies and percentages. Continuous variables between groups were compared with *t*-Student test or Mann–Whitney U test, according to the Kolmogorov–Smirnov test. Categorical and dichotomous variables were compared between groups with chi-square test or Fisher’s exact test according to the sample size. BC: breast cancer; EN: endometriosis; SLNB: sentinel lymph node biopsy; ER: estrogen receptor; PR: progesterone receptor; HER2+: human epidermal growth factor receptor 2; Ki67: Antigen Kiel 67. *p*-values < 0.05 are highlighted with *.

**Table 3 medicina-60-00625-t003:** Multivariate analysis predicting breast cancer with Ki67 index < 20%.

	Overall Population (*n =* 123)			
	OR	Sign.	Wald	95%CI
Age at breast cancer diagnosis	0.816	0.040	4.199	0.671–0.991
Age at menopause	1.925	0.270	0.998	0.532–6.961
Hormonal therapy	0.501	0.107	2.592	0.216–1.162
Endometriosis	3.203	0.012	6.240	1.285–7.986
	Premenopausal Group (*n =* 90)			
	OR	Sign.	Wald	95%CI
Age at breast cancer diagnosis	0.822	0.325	0.969	0.556–1.215
Age at menopause	1.103	0.605	0.267	0.761–1.599
Hormonal therapy	0.568	0.243	1.363	0.220–1.468
Endometriosis	3.882	0.012	6.368	1.354–11.135
	Postmenopausal Group (*n =* 33)			
	OR	Sign.	Wald	95%CI
Age at breast cancer diagnosis	0.803	0.166	1.921	0.589–1.095
Age at menopause	1.138	0.493	0.470	0.787–1.645
Hormonal therapy	0.332	0.252	1.310	0.050–2.197
Endometriosis	1.772	0.556	0.347	0.264–11.884

Variables with *p*-values < 0.10 were included in multivariate logistic regression statistical analysis in order to estimate the effect on less aggressive breast cancer (defined as ki67 < 20%) with the forward Wald method. OR: odds ratio; Sign.: significance; 95%CI: 95% confidence interval.

**Table 4 medicina-60-00625-t004:** Demographic characteristics and pelvic endometriosis side in the premenopausal group with endometriosis and breast cancer and (BC+EN+) and the group without breast cancer (BC-EN+).

	Premenopausal Group (*n* = 90)		
	BC+EN+ Group (*n* = 30)	BC-EN+ Group (*n* = 60)	*p*-Value
Age (y) mean ± SD	42.7 ± 5.0	42.1 ± 5.5	0.617
BMI (kg/m^2^) mean ± SD	22.3 ± 6.0	22.6 ± 4.2	0.778
Age at Menarche (y) mean ± SD	12.2 ± 1.1	12.6 ± 1.6	0.230
Previous hormonal treatment n (%)	23 (77.0%)	48 (80%)	0.743
Infertility yes n (%)	9 (30%)	18 (30%)	1.000
ART yes n (%)	7 (23.0%)	11 (18%)	0.576
Nulliparous yes n (%)	14 (47%)	25 (58%)	0.326
Endometriosis type n (%)			
Endometrioma	16 (53%)	33 (55%)	0.850
Posterior DIE	17 (57%)	51 (85%)	0.003
Anterior DIE	2 (7%)	2 (3%)	0.382
Adenomyosis	13 (43%)	39 (65%)	0.048

All continuous variables are expressed in terms of mean± standard deviation (SD), while categorical variables are indicated as frequencies and percentages. Continuous variables between groups were compared using the *t*-Student test or Mann–Whitney U test, according to the Kolmogorov–Smirnov test. Categorical and dichotomous variables were compared between groups with the chi-square test or Fisher’s exact test, according to the sample size. BC: breast cancer; EN: endometriosis; ART: assisted reproductive technology; DIE: deep infiltrating endometriosis.

## Data Availability

The raw data supporting the conclusions of this article will be made available by the authors upon request.

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
