# Peer review of "Breast Cancer in Patients with Previous Endometriosis Showed Low Aggressive Subtype"

_medicina, 2024, doi:10.3390/medicina60040625_

Round 1
Reviewer 1 Report
Comments and Suggestions for Authors
Dear Authors,
This article focuses on an important topic related to the clinical implications of previous endometriosis in breast cancer. Since the literature has scarce data, this study is essential in identifying the mechanisms that will underlie the best experimental and clinical practices. The authors provided adequate details on methodology, evaluation, findings, and investigations. The particularities and novelty of the article are very well underlined in the results and conclusions sections. Overall, this manuscript is well-written and documented.
However, some suggestions could improve the quality of the article:
● The authors could present the information about participants in a flowchart that will be more attractive to the readers.
● It should be specified the classification of endometriosis (cEnzian and AAGL 2021) in patients who underwent pelvic surgery n=24;
● Why wasn't an additional MRI evaluation performed on these patients to specify the imaging classification?
Kind regards
Author Response
Dear Editor,
We are pleased to submit to Medicina Journal, our revision to the manuscript entitled: “Breast Cancer in patients with previous endometriosis showed Low aggressive subtype”. We are really pleased that both reviewers addressed some improvement that enhanced the quality of our manuscript.
Reviewer #1
- The authors could present the information about participants in a flowchart that will be more attractive to the readers.
Dear Reviewer #1, as requested we presented the data about participants in a flowchart.
- It should be specified the classification of endometriosis (cEnzian and AAGL 2021) in patients who underwent pelvic surgery n=24;
We thank the reviewer for his or her comment. Patients were not all operated in our center but in different Italian centers, each of which uses a different classification for endometriosis staging; for this reason we did not report the surgical classifications, but relied only on the histological report. Of the 21 patients with endometriosis at surgery only 7 had surgery in our University hospital and were classified according #Enzian classification. This is now specified in the result section.
- Why wasn't an additional MRI evaluation performed on these patients to specify the imaging classification?
We thank the reviewer for his or her comment. Since the last 2022 ESHRE guidelines report that both TVS and RMI are accurate diagnostic tools in endometriosis diagnosis, and since nowadays the classifications in use to describe endometriosis have also been applied to ultrasound evaluation (1-2) we choose to use only TVS and we have evaluated patients with clear endometriosis ultrasound signs according to previous published consensus statements (ESUS/IDEA). We now specified this better in the methods section.
Reviewer #2
- Although you included cases with endometriosis-adenomyosis, it must be clarified that they are two similar but not identical conditions.
We thank the reviewer for his or her comment. Patients were not all operated in our center but in different Italian centers, each of which uses a different classification for endometriosis staging; for this reason we did not report the surgical classifications, but relied only on the histological report. Of the 21 patients with endometriosis at surgery only 7 had surgery in our University hospital and were classified according #Enzian classification. This is now specified in the result section.
- It would be interesting to present some relations with endometriosis for women with breast cancer treated with neoadjuvant therapy.
Dear Reviewer #2, due to the immunophenotypic changes related to the neoadjuvant treatment we decided to exclude the patients who underwent to primary systemic treatment. We added this aspect as a limitation in our study and our future studies will focus on locally advanced breast cancer
- In line 501-1, it is written that “Recent studies have established an association between endometriosis and…ovarian cancer. However, this is an “old” knowledge.
Dear Reviewer #2, we corrected the sentence as requested. In fact there are report of this association since several years we change the sentence in:Over the years, several studies have established an association between endometriosis and certain types of malignancies, particularly ovarian cancer.
- In line 65, please correct “oral contraceptives (progesterone, estro-progesterone…) to “oral contraceptives (progestin, estro-progestin…)
Dear Reviewer #2, we corrected the sentence as requested.
- It should be emphasized that mammography is the main method in screening programs (and not ultrasound).
Dear Reviewer #2, we emphasized this aspect in the introduction as requested.
Comments on the Quality of English Language
- There are many errors in the text that should be corrected.
Dear Editor, as requested the error in the text has been corrected.
We would like to extend our sincere gratitude to the reviewers for their invaluable feedback and constructive criticism, which have undoubtedly strengthened our manuscript. We have carefully considered each comment and have made revisions accordingly. We are hopeful that our revisions address the concerns raised and that they will be viewed favorably by the editor. Once again, we thank the reviewers for their time and effort in evaluating our work. We eagerly await the opportunity to share our improved manuscript with the journal.
Thank you for your consideration.
Reviewer 2 Report
Comments and Suggestions for Authors
This is an interesting study. However, some alterations could be made in the manuscript.
1. Although you included cases with endometriosis-adenomyosis, it must be clarified that they are two similar but not identical conditions.
2. It would be interesting to present some relations with endometriosis for women with breast cancer treated with neoadjuvant therapy.
3. In line 501-1, it is written that “Recent studies have established an association between endometriosis and…ovarian cancer. However, this is an “old” knowledge.
4. In line 65, please correct “oral contraceptives (progesterone, estro-progesterone…) to “oral contraceptives (progestin, estro-progestin…)
5. It should be emphasized that mammography is the main method in screening programs (and not ultrasound).
Comments on the Quality of English Language
There are many errors in the text that should be corrected.
Examples:
Line 48: “metastases[3,4]” to “metastases [3,4]”
Line 54: “[7].Nowdays” to “[7]. Nowadays”
Line “73: [8]. In” to “[8]. In”
Line 155: “system.29” to “system [29]”.
Line 170-171: “Adenomyosis[33,34] .” to “Adenomyosis [33,34].”
Author Response

(The authors gave the same response as above.)
